# Safe and Effective Antioxidant: The Biological Mechanism and Potential Pathways of Ergothioneine in the Skin

**DOI:** 10.3390/molecules28041648

**Published:** 2023-02-08

**Authors:** Hui-Min Liu, Wei Tang, Xiao-Yi Wang, Jing-Jing Jiang, Wei Zhang, Wei Wang

**Affiliations:** 1School of Perfume & Aroma and Cosmetics, Shanghai Institute of Technology, Shanghai 201418, China; 2Engineering Research Center of Perfume & Aroma and Cosmetics, Ministry of Education, Shanghai 201418, China; 3Shanghai EGT Synbio Group Co., Ltd., Minhang District, Shanghai 201100, China

**Keywords:** ergothioneine, antioxidant, inflammation, cosmetics, skin

## Abstract

Ergothioneine, a sulfur-containing micromolecular histidine derivative, has attracted increasing attention from scholars since it was confirmed in the human body. In the human body, ergothioneine is transported and accumulated specifically through OCTN-1, especially in the mitochondria and nucleus, suggesting that it can target damaged cells and tissues as an antioxidant. It shows excellent antioxidant, anti-inflammatory effects, and anti-aging properties, and inhibits melanin production. It is a mega antioxidant that may participate in the antioxidant network system and promote the reducing glutathione regeneration cycle. This review summarizes studies on the antioxidant effects of ergothioneine on various free radicals in vitro to date and systematically introduces its biological activities and potential mechanisms, mostly in dermatology. Additionally, the application of ergothioneine in cosmetics is briefly summarized. Lastly, we propose some problems that require solutions to understand the mechanism of action of ergothioneine. We believe that ergothioneine has good prospects in the food and cosmetics industries, and can thus meet some needs of the health and beauty industry.

## 1. Introduction

Ergothioneine (EGT), was isolated and named from rye grain infected with ergot (*Claviceps purpurea*) and other fungi by Tanret in 1909 [1]. Over time, the study of EGT attracted increased attention when large accumulations were found in human blood and liver, and it has been explored for more than a century. EGT is a small molecular substance synthesized by some bacteria and fungi and readily absorbed into the body [2,3,4,5,6]. Additionally, it is an antioxidant with regulatory and protective functions similar to that of vitamins in the body [7,8,9,10]. Many studies have shown that even at very low concentrations, EGT has a curative effect on some diseases, such as acute lung injury [11], steatohepatitis [12], and COVID-19 [13]. However, EGT concentration is not uniformly distributed in the body. Some studies [6,14,15,16] have found that EGT levels in various body parts are often vastly different. Following absorption from the environment, it enters the blood circulation through the expression of the OCTN-1 transporter and travels throughout the body, eventually remaining in certain tissues and organs for several weeks [14]. This is one of the reasons why EGT is higher in blood cells [17]. Most importantly, the accumulation of EGT in the body tends to be targeted; it markedly increases in the areas of injury or inflammation [18]. Therefore, the nature and form of EGT suggest its importance in humans.

Recently, there has been an increased interest in health and beauty. External stimuli such as UV light and air pollution can all cause oxidative damage and skin aging through direct and indirect paths. The specific results are the first signs of aging or skin inflammation. Furthermore, the cosmetics industry has experienced dramatic changes in recent decades, which are manifested in the requirements on efficacy and safety. Therefore, the industry and consumers will likely favor safe and effective raw materials. EGT has the potential to meet this need and has a variety of functions such as antioxidant, anti-aging or anti-UV. Although there are many studies and reports on the pharmacology of EGT, its mechanism and application in the skin have not been comprehensively and systematically summarized. Therefore, the purpose of this review is to understand the mechanistic basis for the cosmeceutical application of EGT. Firstly, we briefly reviewed the structural foundation and properties of EGT. Then we discussed separately the biological activities of EGT, summarizing its possible pathways of action in the skin. Finally, we explored its applications in cosmetics and the future perspectives. It is hoped that the study can help to contribute to a better understanding of the mechanisms of action of EGT and correctly evaluate the potential of EGT as a cosmeceutical.

## 2. The Structural Foundation of Ergothioneine

EGT (2-mercaptohistidine trimethylbetaine) is the trimethylbetaine of L-histidine with a tautomeric thiol/thione group at C-2 of the imidazole ring. However, unlike other water-soluble sulfur-containing derivatives, such as glutathione (GSH), the redox chemistry of EGT is spontaneous and rarely relies on the availability of an enzyme because the thiol tautomer allows the oxidative formation of disulfide bridges—the tautomeric equilibrium [19], which is possibly the reason for the strong autoxidative resistance. In vivo, EGT usually exists mostly in the thione state, ready to play an antioxidant role by capturing radicals (HO^•^, ^1^O_2_^•^, O_2_^•−^, and ^•^NO_2_). Subsequently, EGT is reduced to the thiol state. The regeneration cycle of EGT is tightly related to the GSH/GSSG redox couple, and EGT levels are largely stable in the presence of 10 mM GSH in vitro [20]. Additionally, this is one feasible way EGT uniquely participates in the antioxidant cycle network. In aqueous solution and under physiological pH conditions, EGT-dihydrate has a unique X-ray crystal structure with a carbon-sulfur (C–S) length of 1.69 Å, typical for thiourea [4,21]. This explains its antioxidant properties and ability to chelate metal cations.

## 3. Transport and Metabolism Mechanism of Ergothioneine

Oxidative damage is often a key factor that causes abnormal metabolism, aging, and cell apoptosis. The body repairs damage by building an antioxidant system comprising endogenous antioxidant and repair enzymes [22,23], which synergistically fights against free radicals, protein-carbon-based compounds, and other substances originating from damaged cells or tissues. Furthermore, the system plays a significant role in slowing aging [24,25], counteracting genetic damage [26,27], and enhancing stress resistance [27]. Therefore, a wide range of antioxidants has been discovered and applied to enhance or supplement the antioxidant mechanisms in the human body. Furthermore, due to its unique transport mechanism, EGT is not randomly or uniformly distributed in the various parts of the human body; rather, it tends to accumulate and distribute relatively more in abnormal tissues, during oxidative stress and inflammation, and is abundant during the repair of damaged cells [12]. This indicates that the protective effect of EGT is targeted and specific. This is one of the reasons why EGT is a safe and effective antioxidant.

As a “vitamin”, EGT is not synthesized by the body; rather, it is transported and accumulated in specific parts after ingestion from the diet [28]. Therefore, a specialized and efficient transporter is required, OCTN-1, which is transcribed and expressed by the human gene *SLC22A4* [29]. Although OCTN-1 is a cation transporter, its transport rate of EGT is 100 times higher than that of other transport substrates, such as TEA and carnitine [16]. Additionally, OCTN-1 increases the initial cellular uptake of EGT—which has difficulty penetrating cell membranes—by 600 times [16]. Owing to its hydrophilic nature, EGT does not penetrate the cell membrane; hence, the bi-directional OCTN-1 transport regulates the accumulation and exit of EGT [10]. EGT has strong stability due to the predominance of the thione tautomer at physiological pH and circulates throughout the body in the blood plasma, but OCTN-1 is still required to facilitate uptake by specific tissues, implying that EGT cannot circulate and play a role in the body on its own. Additionally, this explains the experimental phenomenon that the effect of EGT cannot be exerted when OCTN-1 is knocked out [30,31].

The skin is the largest organ in the body [32] and is also where many oxidation reactions occur, causing skin damage and aging. Furthermore, OCTN-1 is expressed in most skin cells, serving to transport and accumulate EGT. Compared with the dermis, the epidermis has a stronger OCTN-1 expression, concentrated in the basal and granular layer cells [33]. These skin cells can proliferate and divide actively, playing a critical role in skin repair and renewal. Moreover, EGT is distributed inside and outside the cell and is particularly concentrated in the interior of the cell’s mitochondria [34]. As shown in Figure 1, it is more susceptible to stress in the mitochondrial DNA (mtDNA) than in the nuclear DNA because the mitochondria have no histones for protection; hence mitochondria lack an efficient DNA repair mechanism compared with the nucleus. Furthermore, the mitochondria are the main sites of cellular energy and metabolism. The electron transport chains generate free radicals and reactive oxygen species (ROS), such as superoxide and hydroxyl radicals, causing redox imbalances. ROS target the mtDNA during this process, leading to DNA nicks, breaks, and mutations, which are also a hotspot in the mtDNA region or D-loop for DNA damage [10]. EGT can rapidly remove abnormally increased ROS in mitochondria to maintain normal levels, directly affecting the activity of key transcriptional regulators involved in stress response, such as SIRTs [35,36], CD38 [37], and PARPs [33,38,39]. These regulators maintain pyridine in skin cells under oxidative stress and nicotinamide adenine dinucleotide phosphate (NADPH) homeostasis, affecting energy and metabolism in skin cells in response to oxidative stress, inflammation, apoptosis, and aging [33]. Lastly, in the nucleus and other parts, EGT can rapidly chelate metal ions, such as iron, copper, and zinc, in the cells without generating additional free radicals through the Fenton reaction [29], thereby preventing cell and DNA damage [40].

## 4. Participation in the Antioxidant Defense System

Hydrophilic antioxidants generally protect soluble proteins, whereas lipophilic antioxidants are better at protecting lipids [10]. Therefore, the body is composed of various antioxidants, forming the antioxidant network system, to cope with diverse types of stress. The common antioxidant defenses (such as vitamin C, vitamin E, glutathione, CoQ-10, and lipoic acid) play an important role in the network, and interact with each other synergistically or via promoting the reduction and regeneration of themselves [41]. Interestingly, EGT is likely to participate in this antioxidant network. Evidence shows that EGT is linked to the GSH/glutathione disulfide (GSSG) redox couple, promoting the cycling reaction, in turn, GSH reverses the oxidation of EGT [20]. Additionally, in the presence of vitamin C, the one-electron-oxidized form of EGT are rapidly reduced by a pseudo-first-order kinetics producing ascorbyl radicals (k = 6.3 × 10^8^ M^−1^·s^−1^) [42]. This is similar to the results of vitamin C repair of the phenoxyl radical derived from vitamin E [43]. Furthermore, vitamin E can significantly enhance the inhibitory effect of EGT on glucose-mediated free-radical-dependent embryo deformity to benefit the management of diabetic embryopathy [44]. However, the interaction of EGT with several other antioxidants remains to be studied; this has great implications for the understanding the roles of EGT in the body and the current antioxidant theory.

On the other hand, to effectively cope with oxidative stress and electrophilic challenges under various environments, a large number of antioxidative enzyme systems exist in cells, including heme oxygenase-1 (HO-1), NADP(H) quinone oxidoreductase (NQO-1), and γ-glutamylcysteine synthetase (γ-GCLC) [45]. These antioxidant enzymes are closely related to the Nrf2/ARE pathway and induce its activation to maintain the efficient operation of the antioxidant system. Interestingly, EGT can activate the Nrf2/ARE pathway and the corresponding pro-inflammatory cytokines, oxidative stress and tumor promoters. Hseu et al. [25] found that EGT significantly promoted Nrf2 translocation and increased nuclear ARE promoter activity in UVA-induced HSF cell models, which caused improvement of the antioxidant enzyme system including HO-1, NQO-1, and y-GCLC.

Furthermore, EGT plays an important role in regulating the AP-1 signaling pathway. AP-1 is a heterogeneous complex composed of Jun, Fos, and ATF family members and is a key regulator of epithelial-to-mesenchymal transition; it plays a critical role in chronic UV irradiation-induced skin cancer development [25]. It was proven that UVA-induced AP-1 (c-Fos and c-Jun) translocation was inhibited by EGT treatment in parallel inhibition of the collagenolytic MMP-1 activation and type I procollagen degradation [25], which is an important basis for the anti-aging effect of EGT.

## 5. Targeted Intelligent Protection to Mitochondria

Since OCTN-1, expressed by the human gene *SLC22A4*, was found in the mitochondrial plasma membrane [14], it is speculated that EGT is a potential mitochondrially targeted antioxidant (MTA). MTAs usually selectively enter the mitochondria in cells at relatively low concentrations, reducing cellular oxidative damage and chronic inflammation caused during hypoxia [46]. Furthermore, although OCTN-1 does not transport EGT alone, it provides a channel for EGT to enter mitochondria with high transport efficiency and specific expression [14,16], which explains (to some extent) as to why EGT can quickly clear mitochondria-specific ROS. Although a study has proven the specific expression of OCTN-1 in the mitochondria of human tissues (brain, muscle, liver, spleen, testis, skin fibroblasts, and lymphoblasts) [34], there is no direct evidence that EGT accumulates “intelligently” in isolated mitochondria. Notably, using isotope labeling, Kawano et al. [14] demonstrated that EGT accumulates significantly in the mitochondria of hepatic cells derived from rats. A reliable way to prove that EGT protects mitochondria is by studying whether the OCTN-1 expression and the change in EGT content during oxidative stress relate to the degree of cell damage and repair. Interestingly, many studies have confirmed the protective effect of EGT on mitochondria, including preventing mtDNA damage [10] and maintaining membrane potential [36], especially when the cells are under stress. Furthermore, many studies have attempted to verify the therapeutic effect of EGT on diseases linked to mitochondrial damage, such as heart disease [47], Alzheimer’s disease [48], and preeclampsia [49] to provide macro evidence for the protective effect of EGT.

## 6. The Biological Effect of Ergothioneine

### 6.1. Antioxidant Effect

EGT presents an advantage regarding the achievement of a strong antioxidant capacity even at very low concentrations (nanomolar) [50]. Additionally, it can scavenge various reactive oxygen and nitrogen species as well as free radical generators, as shown in Table A1, including hydroxyl [51], superoxide [51,52], DPPH [53,54,55], ABTS [53,55], and peroxynitrite radicals [51].

As an essential trace element in the human body, copper and some transition metals, such as zinc and iron, constitute the active centers of various metalloenzymes and play an important role in redox reactions [56,57]. Additionally, some copper is connected to the bases of the DNA and is an important component of chromatin and genetic material [58]. Based on this characteristic, abnormal levels of copper can cause oxidative damage to DNA, proteins, and lipids and induce apoptosis and pathological changes [59,60,61,62]. Furthermore, most components in the antioxidant network undergo a Fenton reaction with copper to generate additional ROS and free radicals, making it difficult to exert the ideal antioxidant effect. Interestingly, there are some reaction mechanisms where EGT can rapidly chelate metal ions, such as iron and copper, to generate stable complexes without the Fenton reaction [29]. Additionally, EGT can compete with other metal complexes, such as histidine, to chelate copper ions, which significantly inhibits the formation of protein carbonyl groups, thereby protecting DNA and proteins from copper-induced oxidative damage [63,64,65].

The specific effect of EGT on the antioxidant network remains unclear; however, it has been shown that EGT can improve the stability of other antioxidant substances and even play a synergistic role in the body. Gregory et al. [66] confirmed the excellent antioxidant properties of EGT using the three antioxidants (EGT, ferulic acid, and GSH) separately and in combination. EGT had stronger antioxidant properties when 50 μM EGT was combined with 100 μM ferulic acid, which prevented oxidative damage and photoaging of skin cells. Furthermore, low EGT concentrations significantly increase the scavenging rate of DPPH free radicals by GSH and nicotinamide. Numerous studies have shown that EGT significantly increases GSH levels, accelerates the GSH and GSSG cycle, and promotes reduced GSH production [67]. Moreover, EGT can support a multi-step non-enzymatic cycle in scavenging singlet oxygen and cooperating with glutathione. This means that elevated levels of GSH allow EGT to regenerate after being consumed by singlet oxygen. Notably, EGT reacts with singlet oxygen at a much higher rate than vitamin C and L-histidine [20]. EGT’s stability and repeated effectiveness are often used as a stabilizer to prevent discoloration and deterioration and have a good antioxidant effect in vitro to delay the aging and death of tissue cells [68,69].

Furthermore, EGT may maintain and regulate NADPH levels. NADPH is an important part of the cellular antioxidant system and an electron source required for fatty acid, steroid, and DNA synthesis [70,71]. However, H_2_O_2_ is generated when NADPH is oxidized to NADP by the NADPH cytochrome c reductase enzyme. In addition, a large amount of ROS is generated under the effect of NADPH oxidases (NOXs) family enzymes, which are important sources of mitochondrial ROS [72]. Ideally, EGT can chelate the metal ions that promote the H_2_O_2_ reaction and directly inhibit the activity of NOXs, thereby reducing the pathway generating ROS [8]. In summary, EGT makes the cellular antioxidant network cycle more rapid, removes metabolic wastes and ROS, and repairs oxidative damage.

### 6.2. Anti-Aging Effect

The skin is the largest organ and outermost barrier of the human body; therefore, skin aging is the most intuitive expression of the influence of age and external factors [73]. The skin aging process is complex but the reasons can be roughly classified into two aspects: exogenous and endogenous aging [74]. The former involves cell damage or premature aging due to environmental factors, such as UV and pollution [75]. In contrast, the latter reflects to that with age, the ability of cellular immune defense mechanisms [76] and self-renewal declines [77]. The exhaustion of stem cells and the weakening of metabolic capabilities [78] lead to the inability to renew skin cells and process the harmful substances produced, ultimately leading to skin aging. During this process, nicotinamide adenine dinucleotide (NADH) and NADPH in the mitochondria play a key role in the cellular antioxidant system, energy metabolism, and electron transfer. Furthermore, mitochondria are responsible for approximately 90% of cellular energy production during the oxidative phosphorylation of NADH, generating large amounts of ROS [79,80]. The body can regulate ROS levels by antioxidant defense systems when there is abnormal accumulation of ROS. However, some ROS circumvent these processes and can damage the mtDNA, proteins, and lipids. Cellular aging can be delayed and repair ability enhanced through these mechanisms [79].

NADH is mainly stored in mitochondrial NADH pools and is involved in cellular energy metabolism [81]. Similarly, OCTN-1 has a higher expression level in the mitochondria, implying that EGT plays a significant role in mitochondria. As shown in Figure 2, EGT can regulate NOXs [82,83], NADH oxidase [31,80,84], and GSH-Px [84]. Therefore, we inferred that OCTN-1 specifically transports EGT into the mitochondria to remove generated intracellular ROS and alleviate the aging process. Additionally, some studies have confirmed that EGT can affect NADH-oxidase [31], promoting NADH conversion to NAD^+^ and storage in the mitochondrial pool, ultimately reducing the NAD^+^/NADH level [85]. This means that EGT reduces the potential production of ROS and activates SIRTs [71,85], PARP-1 [33,71], which are of substantial significance.

As shown in Figure 2, the maintenance and regulation of GSH by EGT can be hypothesized and inferred. According to a study, EGT can inhibit the activity of NADPH oxidase, which reduces the ROS generated from NADPH oxidation and increases the level of NADPH, maintaining the reduced state of GSH [14]. Furthermore, NADPH can maintain GSH in a stable state under the action of GSH-reductase and generate GSSG through GSH-red oxidase, with the participation of EGT to decompose H_2_O_2_. Increase in GSH results in a corresponding increase in the ratio of GSH/GSSG [31]. Pan et al. [83] confirmed that EGT can maintain the ratio of GSH/GSSG and increase lifespan in *Drosophila*.

### 6.3. Anti-Inflammatory Action and Immunomodulation

Oxidative stress or environmental factors in the body often induce immune system stress, triggering inflammatory responses and lesions [86]. Among these factors, exogenous factors, such as UV stimulation [87], harmful gases [88], and trauma [89], are the most common and can trigger inflammatory pathways and activate pro-inflammatory factors, such as IL-1β, IL-6, IL-8, and TNF-α [90]. Besides directly causing cell and tissue damage, these factors can activate other cells in the skin, such as Langerhans cells and lymphocytes [91], to release inflammatory mediators [92], causing local reactions and inducing dermatitis and other skin diseases, including skin cancer [93]. Therefore, it is important to study how to slow down the inflammatory process or block the inflammatory pathway to maintain the skin barrier and health.

As a component of the body’s antioxidant system, EGT has been studied as a potential anti-inflammatory substance, and the results show that it can be used as a backup means for body antioxidation to accumulate and play a role in the site of damage specifically [27,94,95]. Therefore, it is important to understand the role of EGT signal transduction mechanisms in cellular oxidative stress. As shown in Figure 3, EGT is likely to activate the body’s antioxidant repair system by activating the MAPK gene cascade in cells and regulating and controlling cytokines, such as TNF-α, NF-κB, and IL-6 [27,96]. Additionally, it has a corresponding regulatory effect on the activities of MMP-1 [52], NOXs [8], and other inflammatory response-related enzymes [50]. As shown in Figure 1, when the body is subjected to stimuli such as H_2_O_2_, UV, PA, and other factors, OCTN-1 is rapidly expressed or transmitted, and the EGT content in the damaged part is significantly increased [96].

Furthermore, EGT inhibits TNF-α and activates NF-κB by stimulation and induces the protective effect of the ARE/Nrf2 and Hsp70 pathways to activate the antioxidant system to quickly remove and inhibit the production of harmful free radicals and inflammatory factors [50]. Additionally, with EGT accumulation, oxidative damage and inflammation markers, such as allantoin, 8iso-PGF2G, and C-reactive protein, were significantly reduced at the stress site [8]. Interestingly, OCTN-1 expression may lag the occurrence of an inflammatory response, and the gene expression region of OCTN-1 is in chromosome 5q31, which is related to various immune and inflammatory system genes [95]. This may imply that OCTN-1 and EGT are closely related to the immune system and act as a final barrier to damage to the body.

The high expression of OCTN-1 in immune cells also suggests the role of EGT in immune regulation. It has been proved that EGT may be involved in the regulation of the immune system as a Toll-like receptor agonist [97]. EGT acts specifically on macrophages with the participation of OCTN-1 transporter to induce Th17 shift in CD4^+^ T cells. Further, pretreatment with EGT enhances transcription of M1-related cytokine genes induced by TLR ligands, which reduces the expression of IL-10 relative to TLR. It is well known that IL-10 is usually associated with acne inflammatory lesions in the skin and inhibits the activation and proliferation of T cells via inhibition of antigen-presenting cells including Langerhans cells [98]. Furthermore, EGT participates in the inhibition of JNK-mediated IL-6 and TNF-α-mediated IL-8 production [96,99], which has key regulatory roles in immune and inflammatory responses. However, the role of EGT in immune cells may be different from that in other cell types, which is reflected by the fact that EGT as a TLR agonist enhances the expression of IL-1β, IL-6, and IL-12p40. IL-6 and IL-23 regulated IL-12p40 promoted Th17 differentiation in naive CD4^+^ T cells [97]. The degree of Th17 polarization induced the promotion of inflammation-based autoimmunity although the antioxidant effect on the production of IL-6 and IL-12p40 is controversial.

### 6.4. Protective Effect on Skin Barrier

The skin comprises the epidermis, dermis, and subcutaneous tissues, forming a complete barrier. These layers are formed by different cell differentiation and arrangements and constitute the first interface between the body and the environment, as shown in Figure 4. Moreover, the skin forms physical, chemical, and stable biological barriers that can house different microorganisms and maintain ecological balance. Damage to this skin barrier function can lead to skin diseases or infections, such as atopic dermatitis (AD) and acne. Rizzo et al. [100] demonstrated that the p53 family p63 might be key in driving AD. Moreover, p53 affects cell proliferation and differentiation by regulating RUNX1, which is generally detected in the nucleus of epidermal cells but is not significantly expressed in patients with AD [101]. This may suggest that it is related to the reconstruction of the skin structure and the repair of the skin barrier.

Interestingly, OCTN-1 expression was strongly correlated (score = 0.999) with RUNX1 gene expression, reflected in the STITCH database, as shown in Figure 5. Moreover, a study has demonstrated that EGT has a positive effect on immune system diseases caused by abnormal RUNX1 expression such as rheumatoid arthritis (RA) [102,103,104] and non-alcoholic fatty liver disease [12]. Additionally, RUNX1 expression—similar to that of p63 and p53—can directly regulate the process of cell proliferation and differentiation and is also a vital regulatory factor for normal hematopoiesis in the body, which may be consistent with the result that the blood contains high concentrations of EGT [101]. Moreover, it is a key regulator of normal hematopoiesis in the body [105], which may be consistent with the abundant accumulation of EGT in blood [15]. Furthermore, the regulatory effect of RUNX1 on *SLC22A4* and the activation effect of ROS on p53 may suggest that EGT is transported to clear excess ROS through the high expression of OCTN-1 or that the anti-inflammatory effect of inhibiting IL-33 alleviates AD after the skin barrier function is impaired. This may lead to the downregulation of the IL-33-mediated inflammatory markers (loricrin, keratin 1, and keratin 10) [106]. Overall, these findings suggest that EGT may play a significant role in remodeling skin barrier function and treating AD.

### 6.5. Protective Effect on UV Exposure

UV exposure is one of the most important contributors to skin damage and aging [107]. It is well known that the formation of many harmful products causes skin damage from the action of light in cells or tissues under UV stress, such as the formation of ROS, including superoxide anions, singlet oxygen, hydrogen peroxide, and hydroxyl radicals [108] which can induce a series of MAPK pathways and produce inflammation [27,109]. Therefore, EGT, an important part of the body’s antioxidant system, will undoubtedly become a research focus for potential skincare. Additionally, EGT has great potential in sunscreen products because of its absorption spectrum in the UV range, with a molar extinction coefficient of 1.4 × 10^4^ M^−1^·s^−1^ and a maximum wavelength of 257 nm [10], which is similar to that of octyl methoxycinnamate (320 nm), a common sunscreen [110]. The absorption band at 258 nm probably arises from electronic transitions in the thione form of the 2-mercaptoimidazole moiety of EGT which is destroyed by proton loss from nitrogen (pKa 10.8) [111]. This indicates that EGT in the thione state can act as a physiological UV filter to absorb or shield UV rays and produce bioactive effects, which may be one of the reasons why it blocks UV-induced damage. However, the reduction of skin collagen synthesis and matrix metalloproteinase activity caused by UV is a characteristic of photoaging, which causes a series of symptoms such as roughness, wrinkles, sallowness, telangiectasia, depigmentation, and even skin cancer [25]. In vitro, EGT was shown to be a potent inhibitor of AGEs formation and AGEs cross-linking with collagen [24] and protect these cells by reducing MMP-1 degradation and enhancing collagen I production [79]. This means that EGT has many advantages being a natural antioxidant for the skin and an effective anti-photoaging agent, with the potential to be an excellent UV protector for developing skincare products or healthy sunscreens [79,112].

The mechanism of EGT in UV protection is similar to that of antioxidation and the alleviation of inflammation. This means that UV induces and aggravates oxidative or inflammatory damage. However, the difference is that prolonged UV exposure may cause extra DNA damage. It has been shown that the photoaging skin is characterized by increased mutations in the mitochondrial genome [113]. The most commonly reported deletions are large-scale (4977 bp) deletions, also known as “common deletions” (CDs) [114,115]. In human skin, CD occurs more frequently with increased UV exposure, with a 10-fold increase in the “common absence” of photoaged skin compared with the sunscreen used by volunteers [114]. Based on the effect mechanism of EGT and the OCTN-1 distribution, it can be inferred that EGT acts in the mitochondria to inhibit DNA damage. Additionally, a study showed that when 20 µM EGT was co-incubated in fibroblasts exposed to UVA for 3 weeks, no CD was observed, suggesting that EGT prevents this specific damage in mtDNA [79]. Another study demonstrated that EGT (500 nM) pre-treatment effectively prevented UVA-induced loss of TUNEL-positive nuclei and mitochondrial membrane potential [50]. Therefore, EGT may be involved in protecting mitochondrial DNA from negative effects during the electron transport cycle.

### 6.6. Inhibition Effect on Melanin

Melanin in the skin is a natural barrier providing UV protection [116]. It is synthesized by melanocytes in the epidermal/dermal border of the skin and plays a vital role in protecting skin health [117]. However, too much or too little melanin can cause abnormal skin color and skin lesions, such as age spots and chloasma [118,119]. Therefore, skin care products with whitening and spot-removal functions occupy an important position in cosmetics. Recently, many studies have concentrated on the activity of tyrosinase and the changes in the amount of melanin produced by cells or tissues.

The mechanism of action of EGT in inhibiting melanin production or removing melanin is likely due to its unique structure, a sulfur-substituted imidazole ring [120,121]. Thiol-containing compounds are believed to be important melanogenesis inhibitors because they react with dopaquinone to form colorless conjugates [122]. Additionally, EGT is unique in that its sulfur atom exists as a thione, and not as a sulfhydryl group [123]. This means that the thione structure may differentiate the inhibition mechanism of EGT from that of other tyrosinase inhibitors, such as thiol. Moreover, Liao et al. [120] demonstrated that EGT could bind to the substrate site of the free enzyme and a different substrate site of the enzyme-substrate complex. In addition, observation of inhibitor concentrations suggested that EGT did not reduce the amount of enzyme but caused inhibition of enzyme activity, indicating that EGT is a reversible tyrosinase inhibitor [120]. Furthermore, other studies have confirmed the inhibitory effect of EGT on tyrosinase; this inhibitory effect is better than that of common inhibitors, such as kojic acid and arbutin [55,124].

## 7. The Application of Ergothioneine in Cosmetics

In recent years, natural raw materials have attracted extensive attention in cosmetics. There is a growing consumer demand for cosmetics containing natural or organic ingredients that are perceived as healthier, safe and ecological. EGT can be adapted to most formulations due to its excellent efficacy, stability and water-solubility. Furthermore, EGT produced by biosynthesis technology has a higher purity and sustainability than natural extracts. Some of skincare products (available in 2023) containing EGT are summarized in Table A2. EGT has been widely used in nutrition, anti-aging and sunscreen cosmetics and is not limited by a particular formula system. Further, oral forms of nutricosmetics or supplements containing EGT have been on sale, which often interact with other vitamin ingredients. However, high purity EGT or fermentation products are relatively expensive to apply in cosmetics, which suggests that more cost-effective synthetic purification technology is also valuable to study.

## 8. Conclusions and Perspectives

EGT has extensively been discussed and researched by many scholars since its discovery, and its structure and effects are gradually becoming clear. We believe that the protective mechanism of EGT has great potential and warrants further studies. In this review, EGT is shown to regulate ROS directly in cells or mitochondria as an effective antioxidant. Moreover, SIRTs and Nrf2 pathways can be activated to enhance the antioxidant capacity and mitigate oxidative stress. Some scholars have proposed comparing EGT to vitamins, which play an important role at very low concentrations. The action of EGT in absorbing UV and inhibiting melanin generation may be due to the inherent characteristics of its structure. Inhibition of inflammation and activation of immunity by EGT may also be related to the regulation of ROS, or be dominated by AP-1 and MAPK signaling pathway, or both. Further, it remains a question whether there are other undiscovered mechanisms or pathways.

In cosmetics, people are always searching for an efficient raw material that is both natural and safe. Therefore, a substance it will likely be accepted if it is a natural factor derived from the human body. There is no doubt that EGT fully meets these requirements. Furthermore, it is a hot ingredient in skin care because of its safety, unique antioxidant ability, targeted anti-inflammatory ability, source-based anti-aging ability, noncompetitive whitening ability, and excellent stability. However, most existing application research studies EGT’s effect inside the human body, with little focus on the skin layer. Therefore, considering the progress of the existing research on EGT, we propose several directions for problems that have not been solved or have prospects.

What negative effect can EGT deficiency or excess cause to the skin?What is the transdermal osmotic rate of EGT in each skin layer? Can OCTN-1 can promote the penetration and absorption of EGT?What is the central role of EGT in the body’s antioxidant network? What is a possible mechanism of action?Can EGT transport and penetration be enhanced by adding artificially extracted or synthetic OCTN-1?There are few studies on the effect of EGT in inhibiting melanogenesis, and blocking the melanogenesis pathway is not yet clear.Most of the evidence is based on animal experiments or cell experiments in vitro, and human and clinical evidence is lacking, especially evidence for external application on the skin.

## Figures and Tables

**Figure 1 molecules-28-01648-f001:**
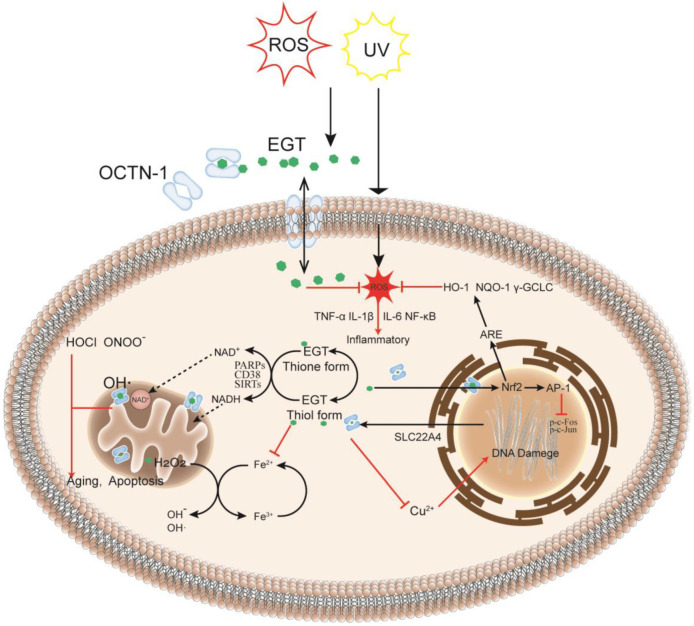
EGT metabolism in the cell: as an intracellular antioxidant, it may have a protective effect on oxidative damage and inflammation caused by exogenous factors reactive oxygen species (ROS) or UV. When cells are damaged or stressed, OCTN-1 transporters are expressed in large quantities to transport EGT across membranes to cells, and EGT can protect cells in different intracellular subcellular organelles in many ways. After entering the cells, EGT can rapidly chelate metal ions in the cytoplasm to prevent further damage to cells by the Fenton reaction. Simultaneously, EGT potentially regulates the activities of PARPs and sirtuin family enzymes and indirectly affects the transport and circulation of NAD+ in the cytoplasm and mitochondria, directly affecting ROS levels in mitochondria. In the nucleus, EGT protects the normal transcription of DNA and prevents deletions, especially DNA damage caused by copper ions. Additionally, EGT can inhibit the production of tumor necrosis factor-alpha (TNF-α), interleukin (IL)−1β, IL-6, and other inflammatory factors, thereby protecting the cells.

**Figure 2 molecules-28-01648-f002:**
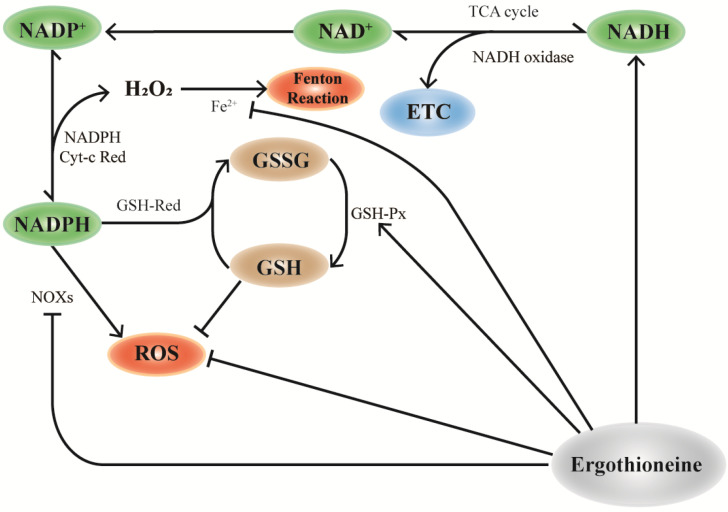
Mechanism of EGT and NAD+ cyclic pathway. In mitochondria, the consumption of NAD+ participates in the TCA cycle of cell energy metabolism and NADP+ production by NAD is influenced by NAD kinase, which further generates NADPH. NADPH can be used as the substrate of glutathione reductase (GR) to reduce GSSG to glutathione (GSH), which is necessary for the activities of the antioxidant enzymes glutathione peroxidase (GPx) and glutathione transferase (GST) and is one of the important ways to improve the antioxidant capacity of cells. EGT promotes this effect by regulating the action of NAD+ pools in mitochondria. Additionally, the antioxidant properties of EGT can directly remove excess ROS, upregulate the activity of GSH-px, and synergically participate in the antioxidant effect of cells.

**Figure 3 molecules-28-01648-f003:**
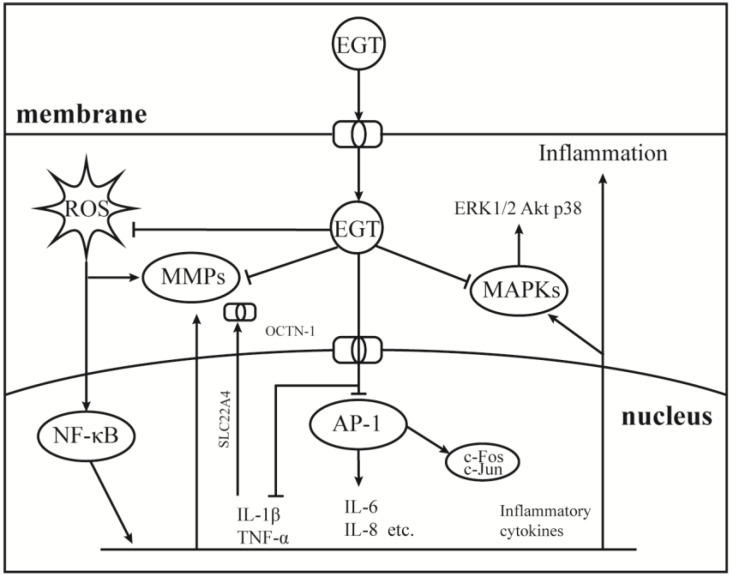
Anti-inflammatory mechanism of EGT on AP-1 and MAPKs signaling pathway. Under the induction of environmental pressure, UV or oxidative damage, ROS can easily attack the nucleus and produce a large number of inflammatory cytokines, then activate the NF-κB cascade signaling pathways. Therefore, the nucleus began to express OCTN-1 in large quantities to promote more EGT transport into cells to play a role. While clearing ROS, EGT inhibited the production of activated AP-1 and MAPKs cascade signaling pathways to relieve inflammation.

**Figure 4 molecules-28-01648-f004:**
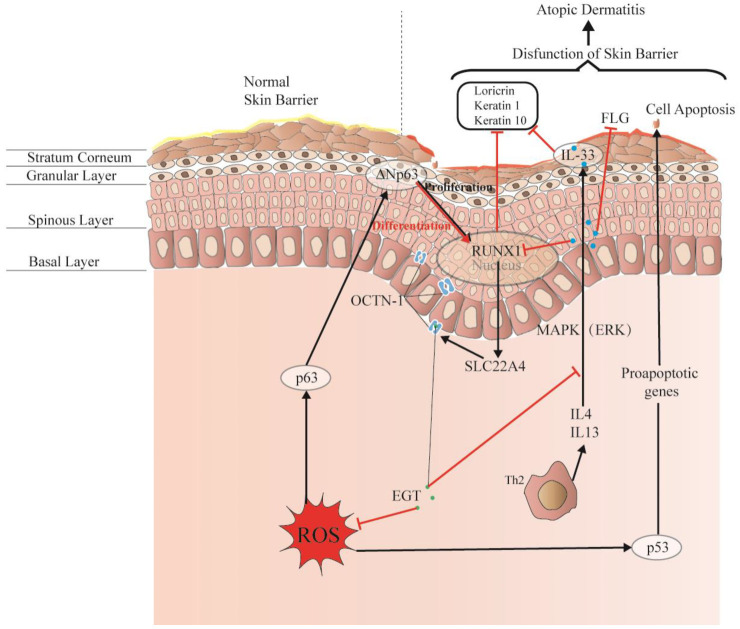
EGT potentially regulates RUNX1 via the protective skin barrier. Additionally, EGT can potentially downregulate the inflammatory factor IL-33 induced by MAPK kinase, reduce the production of inflammatory markers (loricrin, keratin 1, and keratin 10), and indirectly regulate apoptosis caused by p53/63 by inhibiting the excessive increase in ROS. Interestingly, EGT and RUNX1 protein expression induce a strong binding energy related to the proliferation and differentiation of keratinocytes or skin stem cells, and ultimately promote the renewal of skin barrier function.

**Figure 5 molecules-28-01648-f005:**
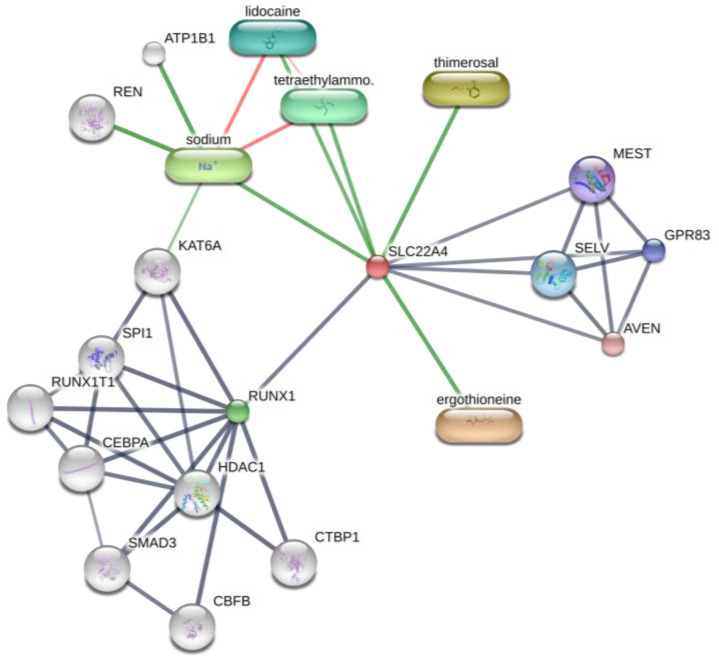
Prediction of EGT binding with related proteins by STITCH database cluster analysis.

## Data Availability

Not applicable.

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
