# Peer review of "Safe and Effective Antioxidant: The Biological Mechanism and Potential Pathways of Ergothioneine in the Skin"

_molecules, 2023, doi:10.3390/molecules28041648_

Round 1

Reviewer 1 Report

In this manuscript the authors review the application of the antioxidant and potential applications in the skin. Thematically this is a much needed review and would be of great interest for the cosmetic and skin care industry. However, the execution of this review is rather poor, and strays away from the title, instead becoming a broad overview of the ergothioneine field (which is not novel, there are a multitude of reviews on ergothioneine already). The layout is rather fragmented and difficult to follow. Most critically there are many sentences that are factually incorrect, some due to poor choice of words. Some references seem incorrect or taken out of context. Examples of these points are given below;

Major issues:

1) Figure 1 is not correct. What does GSH and ROS to do with the tautomeric structure of ergothioneine? The equation is not balanced unless this is at a non-physiological pH which is not indicated. The thione should predominate at physiological conditions.

2) Line 34: Ref 11 doesn't really see ET in a positive light regarding Crohn's disease. This is to do with the polymorphism of the transporter.

3) Line 39: I don't think ref 10 is correct.

4) Problem with the link to the Figures in lines 60, 114, 243, 251, 279, 303, and 315

5) Line 78: ET doesn't only exert effects in 'abnormal' tissues

6) Line 69-70: The body doesn't repair damage but building up antioxidants. Please rephrase this sentence. Also the body's antioxidant defense doesn't simply comprise of 5 antioxidants. There is a huge network of endogenous antioxidant defenses and repair enzymes. Please refrain from taking this "antioxidant network" theory out of context.

7) Line 60-61: Sentence does not make sense. GSH doesn't not participate in conversion of ET from thiol to thione.

8) Line 80: "This indicates that ET is a barrier - the final life of defense for antioxidant mechanisms" - this is a very speculative comment with no factual basis to support this.

9) Line 88-90 - Statements incorrect. ET does not penetrate the cell membrane due to its hydrophilic nature.  OCTN1 transport is bi-directional. The stability of ET is due to the predominance of the thione tautomer at physiological pH. ET does circulate throughout the body in blood plasma but OCTN1 is required to facilitate uptake by specific tissues. Suggest rephrasing these sentences.

10) Line 135: Again these are not the only antioxidants in the body as suggested here. I believe the authors of this reference were only using these as examples of the interlinked nature of antioxidants.

11) Line 156-158: This reference (ref 15) refers to rat liver. There is no "proven specific expression of OCTN1 in mitochondria of human tissues (brain, muscle, liver, spleen, testis,....)" If there is please cite the correct reference.

12) Line 161: I believe it is derived from rats not rabbits.

13) Line 171: "strong AOX capacity even at (nanomolar) concentrations" Please give a reference for this.

14) Line 173-174: These examples are reactive oxygen and nitrogen species and free radical generators, not oxidative damage products

15) Line 227: Mitochondria do not scavenge or inhibit ROS

16) Line 244: "ET can regulate and activate various enzymes in mitochondria" Please provide references.

17) Line 390: Discovery not advent.

18) Line 398: Point 2 - sentence doesn't make sense and did you mean in vivo?

19) Line 406: What "various effects" are you referring to?

20) Line 408: No need for conclusion - suggest delete.

Minor issues:

1) Please improve clarity of Figure 3.

2) Line 339: Perhaps discuss whether most effects seen in UV protection of the skin is mediated by absorption by ET in the UV spectrum

Reviewer 2 Report

The subject matter of the article is in line with current trends in the cosmetics market regarding the search for safe and effective ingredients for this type of formulation. The article summarizes information on ergothioneine and its use in the food and cosmetic industries. The authors paid special attention to the biological action of ergothioneine, which is not widely described in the literature. Therefore, taking up the present topic is justified and contains elements of novelty.

Nevertheless, in the course of reading, the following shortcomings were noted:

- I suggest replacing the word "mega" in the title of the article, such as Safe and effective antioxidant

- In the reviewer's opinion, the introduction is too general. In particular, it should be expanded to include the use of ergothioneine in the cosmetic industry. Have such studies already been conducted? What results have been obtained?

- 2. the structural foundation of ergothionein - it would be good to give here the exact chemical name of ergothionein

- l.114 p. 4- Error! Reference source 114 not found

- L.243 p. 7- Error! Reference source 114 not found

- l.251 p. 7- Error! Reference source 114 not found

- l.279 p. 7- Error! Reference source 114 not found

- l.303 p. 8- Error! Reference source 114 not found.

- l.315 p. 9- Error! Reference source 114 not found.

- Discussion- is too general, especially the first part of it. You should write specific conclusions based on the literature stage prepared by the Authors

Reviewer 3 Report

The following query should be addressed properly 

1.       There are some typographical mistakes. For example. Line 60, 114, 243, 251, 279, 303-304, 315 

2.       How Ergothioneine regulates the Nrf2-mediated antioxidant effect is not discussed. Please address this point more scientifically. The following article will help: https://doi.org/10.1155/2020/2576823

3.       Graphical representation of the anti-inflammatory effect of Ergothioneine is needed.

4.       The effect of Ergothioneine on the regulation of AP-1 signaling is missing.

5.       Is Ergothioneine is TLR agonist? Need critical discussion. This paper will help doi:10.1371/journal.pone.0169360. 

Round 2

Reviewer 1 Report

Thank you for addressing the comments and concerns. The manuscript is significantly improved over the previous version and reads much better.

Have just a few further minor corrections/comments:

Comments:

Line 19-20: “Additionally, the potential direction or pathway is analyzed and predicted to maintain the skin barrier.” Suggest rephrase as this doesn’t really make sense.

Line 21-23: “We believe that ergothioneine can satisfy people's pursuit of health and beauty and has good prospects in the food and cosmetics industries”. Did you mean meet the needs of the health and beauty industry? Suggest rephrasing this sentence.

Line 31: “synthesised by SOME bacteria AND fungi, and READILY absorbed” – not all bacteria or yeast produce ET.

Line 34: suggest change “curative effect” with “beneficial” or “therapeutic” effects. ET does no completely cure the conditions but may help reduce pathology.

Line 39: “This is one of the reasons why ET is higher in blood cells” – Not sure what this means but it is likely factually incorrect. Higher ET in blood cells would be due to OCTN1 expression as is the distribution in the body.

Line 229: Suggest change “differently and jointly” with “separately and in combination”

Line 265: “inhibit ROS produced by antioxidant defense system” – Did you mean the antioxidant defense scavenges ROS? At the moment it reads as if the ROS is produced by antioxidants.

Line 284: Good to mention examples of those enzymes from the references.

Line 297: “this ratio’s effect” - not the ratio the elevation of reduced glutathione.

Line 379 & 387: Switching between the abbrev ET and EGT for ergothioneine.

Line 462: What is the meaning of “not limited by dosage form”?

Line 471: “we believe that the PROTECTIVE mechanisms of ET”?

Reviewer 3 Report

The authors provide my all query quite perfectly. Now the present form of the manuscript is far better than the previous one. The present form of the manuscript is now considered for publication. 

Author Response

Thank you very much for taking time to review this manuscript again.